# Alleviating Forgetfulness of Linear Attention by Hybrid Sparse Attention and Contextualized Learnable Token Eviction

## Abstract

Linear-attention models that compress the entire input sequence into a fixed-size recurrent state offer an efficient alternative to Transformers, but their finite memory induces forgetfulness that harms retrieval-intensive tasks. To mitigate the issue, we explore a series of hybrid models that restore direct access to past tokens. We interleave token mixers with intermediate time and space complexity between linear and full attention, including sparse attention with token eviction, and the query-aware native sparse attention. Particularly, we propose a novel learnable token eviction approach. Combined with sliding-window attention, an end-to-end trainable lightweight CNN aggregates information from both past and future adjacent tokens to adaptively retain a limited set of critical KV-pairs per head, maintaining linear attention's constant time and space complexity. Efficient Triton kernels for the sparse attention mechanisms are provided. Empirical evaluations on retrieval-intensive benchmarks support the effectiveness of our approaches.

## 1 Introduction

The recent emergence of large language models (LLMs) highlights the strong capability and efficiency of the transformer architecture (Vaswani et al., 2017). By explicitly scoring pairwise token relevance, transformers excel in capturing long-term dependence and retrieving information from the distant past. Furthermore, transformers can be efficiently trained in parallel on an unprecedented scale of model and data capacity on modern GPUs. All these advantages enable transformers to supplant recurrent neural networks (RNNs) in the field of natural language processing. However, the blessing of pairwise attention is also a curse: compute time scales linearly with past context length per token (i.e. quadratically for the whole sequence), and efficient decoding requires a key-value (KV) cache that grows linearly in GPU memory; both of which drive computational costs and hardware demands. In response, novel hardware-efficient recurrent or linear attention token mixers such as Mamba (Gu & Dao, 2024) and DeltaNet (Yang et al., 2024d) have gained traction. Like early RNNs, these models compress the entire history into a fixed-size memory recurrently updated for each new token (illustrated in Figure 1 (a)), yielding $O(1)$ time and space per step, while remaining competitive with transformers on many language tasks. However, this also comes with a cost: a variable-length history can never be losslessly compressed into a fixed-size state. As time elapses, information from distant tokens inevitably decays, resulting in diminishing performance on retrieval-intensive tasks compared to standard transformers, even at modest context lengths (Wen et al., 2025; Jelassi et al., 2024; Arora et al., 2024a;b).

Various methods have been proposed to mitigate this forgetfulness. A prominent direction is to improve the update rule itself to increase memory expressiveness and update selectivity (Gu & Dao, 2024; Yang et al., 2024b; Du et al., 2025), without altering the recurrent nature. Others adopt hybrid models by interleaving full and linear attention layers (Lenz et al., 2025; Waleffe et al., 2024; Mini-Max et al., 2025). While this combination or compromise of two worlds allows more effective direct retrieval of past tokens, the complexity issue resurfaces. This motivates a natural question: Can we directly access past tokens for better retrieval without much sacrifice in computational complexity?

Other than a complete fall back from linear to standard attention, there exists a series of token mixers with time and space complexity between the two extremes, as exemplified in Figure 1. Among them

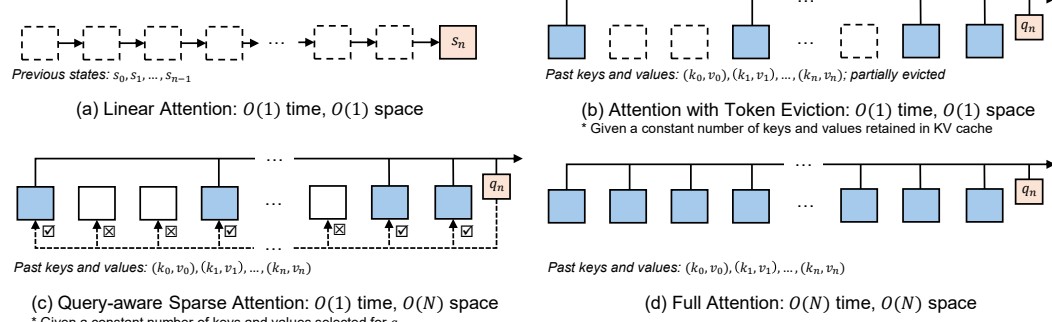

Figure 1: Hierarchy of token mixers across different time and space complexity per step. More complete and direct access to past tokens entails higher time and space costs.

recent query-aware sparse attention mechanisms such as MoBA (Lu et al., 2025) and Native Sparse Attention (NSA) (Yuan et al., 2025) are promising options. These models introduce a lightweight "probing" step that first compares the current query with blocks of previous KVs. Only a small set of possible most relevant past tokens are selected for the actual attention computation, so as to approximate full attention without $O(N)$-time computation. This motivates our proposal of a hybrid **laNSA** model that interleaves **l**inear **a**ttention and **NSA** layers, aiming at directly accessing past information while largely preserving the favorable time complexity.

However, these methods still require accessing all past tokens and thus an $O(N)$-size KV cache, undermining linear attention's constant-space appeal; block-wise probing also limits granularity and adds overheads. Ideally, one could determine or *forecast* if the current token will be relevant for future queries using certain relevant criteria, and proactively discard less relevant tokens in advance, retaining only $O(1)$ tokens in GPU memory (Figure 1 (b)). This has inspired a thread of research from the earlier sparse and local attention (Beltagy et al., 2020) to recent token eviction rules driven by accumulated attention scores (Zhang et al., 2023; Chen et al., 2024b). This approach yet leads back to forgetfulness: despite the direct access to the distant past, once a token is dropped, it is no longer available to any future query. An accurate eviction rule is henceforth of utmost importance. Most prior work relies on fixed, handcrafted heuristics that are not input-adaptive. By contrast, we introduce learnable token eviction (LTE), which leverages a novel lightweight convolution neural network (CNN) module that combines sliding-window attention (SWA) to predict the individual KV importance per head from both the past and future adjacent context. Trained end-to-end under sparsity regularization, LTE allows each sparse attention layer to retain the most relevant KVs under a learned per-head cache budget. We then build our **laLTE** model with interleaved linear and LTE-sparse attention layers. We further approach actual speedup with a dedicated KV cache layout, fused Triton kernels for SWA+KV-sparse attention computation and caching, and a lazy scoring scheme. In this way, laLTE retains both the constant time and space complexity of linear attention, while aimed at best approaching the performance of laNSA and standard transformer models.

We conduct a series of empirical studies to evaluate our approaches at the scale of 0.4B and 1.4B models on several short-context and long-context/retrieval-intensive benchmarks. Both the laLTE and laNSA models outperform baselines including hybrids with SWA or heuristic eviction, and approach the performance of full transformers. To summarize, our contributions are threefold: 1) We propose a novel contextualized learnable token eviction mechanism; 2) We build hybrid laNSA and laLTE architectures with efficient implementation; 3) We examine the performance of hybrid linear attention models across the token mixer hierarchy. Our source code and models are available at [ANONYMIZED].

## 2 BACKGROUND AND RELATED WORK

### 2.1 SPARSE ATTENTION

**Sparsity patterns** A substantial body of work is aimed at reducing the $O(N^2)$ time complexity of transformer stemmed from the attention computation $A(\boldsymbol{Q}, \boldsymbol{K}, \boldsymbol{V}) = \text{softmax}(\frac{\boldsymbol{Q}\boldsymbol{K}^T}{\sqrt{d}})\boldsymbol{V}$ or $A(\boldsymbol{q}_i, \boldsymbol{K}, \boldsymbol{V})$ for each query vector $q_i$. The $N \times d$ matrices $\boldsymbol{Q}$, $\boldsymbol{K}$, and $\boldsymbol{V}$ represent the sequence

of $N$ $d$-dimensional *query*, *key*, and *value* vectors. A common approach is sparse attention, i.e., limiting the attention computation of each query to a small set of KV pairs $\boldsymbol{K}_{\mathcal{I}}$, $\boldsymbol{V}_{\mathcal{I}}$ at indices $\mathcal{I} \subseteq \{1, 2, ..., n\}$ (or $\mathcal{I}_i$ per $q_i$) of tokens believed to be important, reducing the overall complexity to $O(N|\mathcal{I}|)$. $\mathcal{I}_i$ can be simply determined by fixed patterns, e.g., by selecting special tokens (Child et al., 2019; Ge et al., 2023) or specific positions. A prominent example is sliding-window attention (SWA), that selects the most recent $w$ tokens and discards KVs that fall outside the window during decoding. SWA is often combined with "global tokens", e.g., the first $s$ tokens of the input (Beltagy et al., 2020; Ainslie et al., 2020; Zaheer et al., 2020), a.k.a. attention sink (Xiao et al., 2023). The resulting A-shaped pattern $\mathcal{I}_i = \{j : j \le s \lor i - w < j \le i\}$ helps a pretrained standard transformer to generalize to longer contexts (Han et al., 2024). Despite simplicity and efficiency, fixed patterns are nevertheless agnostic to the encoded inputs and show suboptimal performance.

**Token eviction**   Many works focus on more flexible strategies to identify tokens believed to be important for future attention, discarding less important $k_j, v_j$ from the KV cache or evict them from any future $\mathcal{I}_i, i > j$ to maintain a token budget $|\mathcal{I}_j| \le S$. Because high historical attention score (i.e. more contribution to past outputs) often correlates with future importance, a common strategy retains tokens with top-K accumulated attention scores in the whole history (Wang et al., 2021; Zhang et al., 2023; Jo & Shin, 2024), in a window (Liu et al., 2023; Zhao et al., 2024; Oren et al., 2024; Li et al., 2024), or to specific query tokens (Chen et al., 2024b; Kim et al., 2024). Notably, attention patterns vary by layer and head, which motivates per-head cache budget, possibly with global budget allocation (Feng et al., 2025; Zhou et al., 2025), more budget in lower layers (Yang et al., 2024a; Cai et al., 2024), or to heads with more dispersed attention (Qin et al., 2024b). All these methods are primarily based on heuristics on attention scores and training-free. However, many tokens are *typically* important (e.g., infobox in a Wikipedia article) but without high local importance and attention score. Also, attention scores are not directly available from standard Flash Attention (Dao et al., 2022); extracting them adds implementation complexity and overheads.

A more flexible and potent alternative is to predict *typically* important tokens, possibly for each head, with a learnable module, a direction explored less extensively. Zhang et al. (2024) train a separate RNN to predict token attention scores, without considering any interaction with the base model. Other works co-train LTE together with the whole model. An early attempt is ColT5, which selects tokens with a lightweight MLP router (Ainslie et al., 2023b), taking each token as input and hence not contexualized. Anagnostidis et al. (2023) also train a layerwise bias on attention scores for token importance. Zeng et al. (2024) train a separate attention layer for LTE. Although being fully contextualized, this involves heavy quadratic computation and is limited to prefilling with global context. More aggressive methods skip entire attention layers for tokens selected by per-token routers (Raposo et al., 2024; Jiang et al., 2024b; Sharma et al., 2025), precluding any interaction with skipped tokens. Very recently, several concurrent works explore LTE with per-token single-layer projection: Łańcucki et al. (2025) fine-tune LLMs with a global compression ratio constraint, which allows many layers to remain uncompressed and contradicts our $O(1)$ complexity goal. Piękos et al. (2025) train small scale LMs under predetermined cache budget, without considering layer and head variations. Shi et al. (2025) predict importance scores that are then multiplied with attention scores, which deviates from standard attention computation and leads to extra implementation complexity. As far as we are aware, this paper is the first to introduce a contextualized lightweight LTE variant.

**Query-aware selection**   The token eviction mechanisms above attempt to identify tokens relevant to future queries without actually knowing them; this kind of forecast is inherently fallible. If we relax the space constraint and retain all past KVs, we can select the $\mathcal{I}_i = \text{select}(\boldsymbol{q}_i, \boldsymbol{K}, \boldsymbol{V})$ in a $\boldsymbol{q}_i$-aware manner, via an $O(N)$ but lightweight probing step before the actual attention computation. Early selection methods utilize online hashing and clustering (Kitaev et al., 2020; Roy et al., 2021), while many recent approaches highlight GPU-efficient blockwise approximation, exemplified by Mixture of Block Attention (MoBA) (Lu et al., 2025): input sequences are split into $M$-sized blocks, and keys in each block $B_m$ are compressed into a block-level key $\boldsymbol{k}_m^B$ by

$$\boldsymbol{k}_m^B = \phi(\boldsymbol{k}_j), j \in B_m \tag{1}$$

where $\phi$ denotes mean-pooling. Then $\text{score}(\boldsymbol{q}, B_m) = \langle \boldsymbol{q}, \boldsymbol{k}_m^B \rangle$ represents an approximated attention score at block level, and KVs in $\mathcal{I}_i^{(slc)} = \cup B_m$ for $m \in \text{top-K}(\text{score}(\boldsymbol{q}_i, B_m))$ are used to produce selective attention results $A_{slc}$. Intuitively, tokens in blocks with high attention score between $\boldsymbol{q}_i$ and mean-pooled keys are selected, and the whole network is then trained end-to-end.

In this way, the probing step can be carried out by only $N/M$ attention computations. Given the probing results $\mathcal{I}_i^{(slc)}$, with only a small constant number $MK$ of tokens attended, the actual attention time complexity is kept constant. We further adopt the stronger and more sophisticated Native Sparse Attention (NSA) (Yuan et al., 2025). NSA uses a learned MLP as $\phi$ to produce compressed keys $\boldsymbol{k}_m^B$ and values $\boldsymbol{v}_m^B$. In addition to $A_{slc}$, NSA further computes a compressive attention branch $A_{cmp} = A(\boldsymbol{q}_i, \boldsymbol{k}_m^B, \boldsymbol{v}_m^B)$ directly using the compressed KVs, as well as an SWA branch $A_{swa}$. It then predicts three gating weights $g_{\{slc,cmp,swa\}}$ for each query, and outputs

$$A_{NSA} = g_{slc} \cdot A_{slc} + g_{cmp} \cdot A_{cmp} + g_{swa} \cdot A_{swa}. \tag{2}$$

Other learned variants directly predict block- or token-level attention scores (Gao et al., 2025; DeepSeek-AI, 2025) via a lightweight scoring branch, approximate attention scores in lower rank (Singhania et al., 2024; Tan et al., 2025), combine the NSA branches (Zhao et al., 2025), or use a "landmark" token per block as $\boldsymbol{k}^B$ and $\boldsymbol{v}^B$ (Mohtashami & Jaggi, 2023). Many other methods focus on training-free scenario, and heuristically take block statistics such as mean-pooling as $\boldsymbol{k}^B$ (Tang et al., 2024; Chen et al., 2024a; Jiang et al., 2024a; Xiao et al., 2024; Lu et al., 2024; Zhang et al., 2025; Lee et al., 2024); while we focus on exploring the trained scenario to best fit the model to sparse context.

## 2.2 LINEAR ATTENTION

The recent revival of recurrent models beginning with Linear Attention (Katharopoulos et al., 2020) highlights an efficient alternative to transformers with limited performance degradation. Although both compress past information into a fixed-sized memory, such modern variants differ from classical RNNs in that training can be parallelized as there is no nonlinear transform in the timewise memory transition. Concretely, given $\boldsymbol{q}, \boldsymbol{k}, \boldsymbol{v}$, the output of each Linear Attention step is roughly

$$\boldsymbol{o}_t = \sum_{j=1}^{t} (\boldsymbol{q}_t^T \boldsymbol{k}_j) \boldsymbol{v}_j = (\sum_{j=1}^{t} \boldsymbol{v}_j \boldsymbol{k}_j^T) q_t = \boldsymbol{S}_t \boldsymbol{q}_t, \tag{3}$$

where a matrix-valued hidden state is conceptually updated by accumulating past KVs, i.e., $\boldsymbol{S}_t = \boldsymbol{S}_{t-1} + \boldsymbol{v}_t \boldsymbol{k}_t^T$. More sophisticated update rules have been further introduced, e.g. exponential decay of $S_t$ in RWKV (Peng et al., 2023) and RetNet (Sun et al., 2023), or the parameterized input transforms in S4 (Gu et al., 2022). In these models, past information will be gradually overwritten without explicit mechanisms to decide whether the current input should be stored. Later models including Mamba (Gu & Dao, 2024), Mamba2 (Dao & Gu, 2024), and GLA (Yang et al., 2024c) use input-dependent gating to ignore unimportant inputs and better retain critical past memory. Particularly, DeltaNet improves Mamba by delta rule updates (Schlag et al., 2021; Yang et al., 2024d), and Gated DeltaNet adds additional gating and achieves strong long-context retrieval performance, hence we choose it as a strong baseline and the model backbone in our study. In Gated DeltaNet, the memory update is performed as

$$\boldsymbol{S}_t = \boldsymbol{S}_{t-1}(\alpha_t(\boldsymbol{I} - \beta_t \boldsymbol{k}_t \boldsymbol{k}_t^T)) + \beta_t \boldsymbol{v}_t \boldsymbol{k}_t^T, \tag{4}$$

where the input-dependent gates $\alpha_t, \beta_t$ control the update and retention of past memory. Beyond gating for better selectivity, other works focus on boosting expressiveness of limited memory via improved memory parameterization (Qin et al., 2024a; Peng et al., 2025) or a mixture of routed memories (Du et al., 2025). However, without direct access to past tokens, forgetfulness is inevitable for recurrent models with limited memory.

## 2.3 HYBRID MODELS

To alleviate the forgetfulness of linear attention, a straightforward remedy is to put attention back. Adding a small number of full attention layers can significantly improve retrieval performance (Lenz et al., 2025; Waleffe et al., 2024; MiniMax et al., 2025; Wang et al., 2025) but at increased computational demands. By contrast, we prioritize preserving time and space complexity by interleaving token mixers of intermediate costs. SE-Attn explores parameter efficient fine-tuning of attention layers in hybrid Mamba2 into query-aware sparse attention (Nunez et al., 2024), and laNSA utilizes the more sophisticated token mixer in a different pretraining scenario. Furthermore, it has been observed that interleaving SWA layers can markedly enhance linear attention performance (Ren et al.,

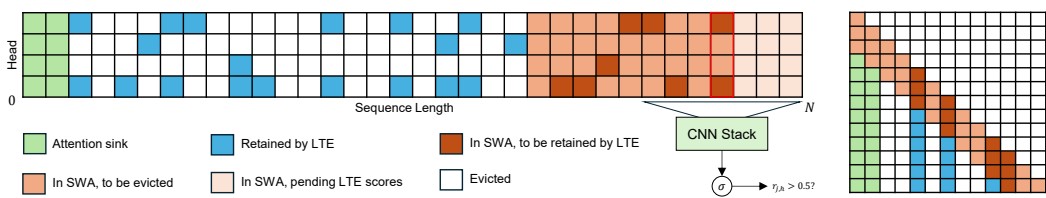

Figure 2: Left: Illustration of our KV eviction scheme, combining LTE, an attention sink (size $s = 2$), and SWA ($w = 12$), with 4 KV heads. A CNN stack is trained end-to-end to predict a per-token, per-head retention score $r_{j,h}$ to decide whether a KV-pair will be evicted when moved out of SWA. With recent KVs cached by SWA, the CNN can read short-range past and future context. $r_j$ can be computed once the CNN receptive field is fully covered by SWA. Right: Resulting per-head A-plus-column sparsity pattern; colored tiles indicate selected tokens at each step.

2024; Zuo et al., 2024; Yang et al., 2024b; Arora et al., 2024b; Cabannes et al., 2025). Closer to laLTE, Ren et al. (2023) and Nguyen et al. (2024) study hybrid models that learn to select tokens as queries to perform SWA. These works nevertheless pursue aims distinct from ours: While SWA can be viewed as an eviction policy, it does not grant access to distant tokens, even when stacked together (Xiao, 2025). Most similar to our work, the concurrent HAXA

## 3 METHODS

### 3.1 LEARNABLE TOKEN EVICTION

#### 3.1.1 LTE MODULE

We retain KV pairs using a per-token, per-head retention score $r_{j,h}$ predicted by the LTE module and restrict attention by combining LTE with an attention sink and sliding-window attention (SWA). Concretely, for the decoding step $i$ and head $h$, the index set is

$$\mathcal{I}_{i,h} = \{j : j \le i, (j \le s) \lor (r_{j,h} > 0.5) \lor (j \ge i - w)\}, \tag{5}$$

which induces an A+column-shaped attention pattern as shown in Figure 2. To determine retention accurately and efficiently, we anchor our LTE module design around four key properties:

1. Fine-grained: Patterns of attention maps vary significantly across layers and heads, some have sharp focus on a few tokens, some demonstrate recognizable local clusters, while others are rather dispersed. Therefore, instead of blockwise and layerwise decision, we choose to make an eviction decision of each KV pair on each head to capture head-specific patterns and make the best use of each head's own cache budget.

2. Contextualized: In addition to each token's own KV, we try to leverage local context including both past and future tokens to better understand each token and its eventual utility. Information from future tokens can not be captured by the causal/recurrent layers used in our model. Hence we use a 3-layer 1D CNN with kernel size=3, dilation=2, and a total receptive field $R = 13$ tokens to ingest local context from both sides. This "look-ahead" is made possible by SWA during decoding: recent KVs are stored in a window $w \gg R$ (e.g., with $w = 768$). We can defer computing $r_j$ by $R//2$ steps until all necessary KVs within the receptive field $[j - R//2, j + R//2]$ are readily available. This computation can be even further deferred, since $r_j$ is not needed as long as token $j$ remains inside the window.

3. Independent: Inspired by ReMoE (Wang et al., 2024), we decide whether a token is important per se, independently without global scheduling such as picking the top-K tokens per head. This not only reduces computational overheads, but leads to simplicity, flexibility, and fine-grained budgeting compared to pre-allocating cache budget $K$ per head base on certain heuristics in other sparse attention approaches described above.

4. Lightweight: We prioritize efficient and parallel inference, hence we choose to apply a small CNN module with only 3 layers and roughly 1% extra parameters in our experiments, and with the channel width halved at each layer.

Figure 2 sketches the full eviction scheme. More specifically, the LTE module consumes the concatenation of key and value vectors per head, prior to rotary positional encodings (RoPE) (Su et al., 2024) for better position generalization. Convolutions are applied independently per head, but executed in parallel across heads using grouped 1D convolutions. Each convolution is followed by Swish activation and a dropout, and a final linear layer maps the per-head feature to scalar scores, also implemented by grouped convolution. Unlike ReMoE, we pass the scalar through a sigmoid to obtain $r_{j,h}$, which stabilizes training. We then binarize the scores with a 0.5 decision threshold. Inputs are zero-padded on the left, and thanks to the deferred computation with SWA, padding is not necessary on the right. The first $s = 4$ tokens are further forcefully retained as the attention sink. We can then simply leverage FlexAttention (Dong et al., 2024) for efficient attention computation with token and head masking during training.

### 3.1.2 ADAPTIVE TRAINING

The model training is a challenge as LTE outputs only $r$ scores to decide a discrete attention mask, hence the gradient cannot be directly back-propagated into LTE parameters. Instead, we leverage straight-through estimator by conceptually applying an all-one mask $m$ to the value vectors so that $v'_{j,h} := v_{j,h} \cdot m_{j,h}$, and use gradients on $m_{j,h}$ as the surrogate gradient, That is to say,

$$\frac{\partial \mathcal{L}}{\partial r_{j,h}} := \frac{\partial \mathcal{L}}{\partial m_{j,h}} = \left\langle \boldsymbol{v}_{j,h}, \ \frac{\partial \mathcal{L}}{\partial \boldsymbol{v}_{j,h}} \right\rangle, \tag{6}$$

In this way, the gradient on $r_{j,h}$ is tied with the contribution to the loss of the underlying $v_{j,h}$ it controls. Stable end-to-end training is observed in practice under this scheme.

Some heads mainly focus on a few tokens, and their attention patterns are naturally sparsified by LTE without any additional constraints: On unimportant $v_{j,h}$, the all-one $m_{j,h}$ is pulled to zero by a negative gradient, which is copied to $r_{j,h}$. Some other heads have a dispersed attention pattern and benefit from an explicit sparsity prior to avoid excessive density. Inspired by ReMoE (Wang et al., 2024), we impose a dynamically adjusted L1-style penalty to encourage sparsity on retained tokens:

$$\mathcal{L}_{sparse} = \sum_h \lambda_h \sum_i \mathrm{ReLU}(r_{i,h} - 0.5) \tag{7}$$

We target a cap $b = 512$ on the number of retained tokens, anticipating $c_h = \sum_i \mathbf{1}[r_{i,h} > 0.5] \leq b$. To adapt the model to this cap, we adjust $\lambda_h$ with a feedback loop during training: increase when the moving average of $c_h$ exceeds $b$, decrease when it falls below $0.95 \cdot b$, with an upper bound $\lambda_h \leq 1$ for stability. Empirically, many heads quickly drive $\lambda_h$ toward zero, suggesting that LTE gradient signals alone suffice to induces adequate sparsity. In this way, the attention density of each head is determined from end-to-end training using gradient signals on the values, without the need of handcrafted rules for budget allocation. More details are described in Appendix A.

### 3.1.3 INFERENCE IMPLEMENTATION

**Capped capacity** Our models are trained on fixed-length sequences ($N = 4096$ in our experiments). Although the number of retained tokens is regularized by the sparsity penalty, the actual computation depends on the retention score of each KV predicted from the input, and no hard cap is enforced as mentioned above. The case is different during inference as we expect guaranteed time and space complexity given arbitrary input with arbitrary length. Therefore we still have to resort to a global budget cap: the number of cached KVs out of window is restricted to at most $b$ per head.

**Cached prefilling and decoding** We first allocate and fill a fixed-size cache as shown in Figure 3 during prefilling, consisting of two segments: one holds the last $w$ KVs within the SWA window, and one gathers KVs already out-of-window but retained by LTE (or as the attention sink). This yields contiguous KVs amenable to tiled attention implementation, avoids memory management intricacies due to cache length imbalance between heads, and prevents worst-case regressions when a head is unusually dense. More specifically, since we store post-RoPE KVs, the physical locations in the cache are irrelevant and we can freely reorder or compact them. This allows us to use the SWA segment as a circular buffer with a `next_ptr` maintained, and compact out-of-window KVs into a contiguous span. Similar to Pagliardini et al. (2023), we leverage this compaction to achieve faster

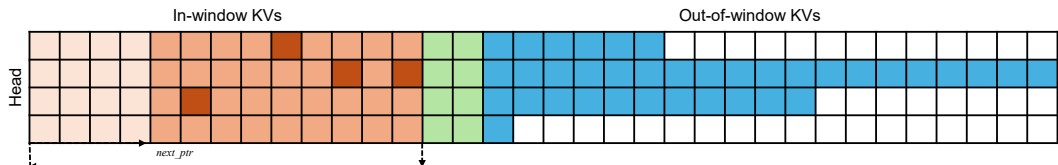

Figure 3: Illustration of the KV cache layout in LTE inference, coloring identical to Figure 2: a circular SWA segment for in-window KVs, and an out-of-window segment with capped capacity for LTE-retained KVs and the attention sink. LTE scoring is batched and deferred as late as possible.

KV-sparse attention. When the capacity happens to exceed on certain head, we keep those with top-$b$ scores. We then implement a Triton kernel for SWA+KV-sparse attention in the tiled Flash Attention manner. Attention outputs of each query tile is computed in two stages: 1) SWA using tiles from original KVs; and 2) KV-sparse attention using tiles from the compacted KV cache. Compacted KV entries inside the sliding window of each query are masked out to avoid double-counting, and tiles completely inside the window (or breaking causality) are skipped for efficiency. During decoding, when a new KV pair is pushed into the SWA buffer, the leftmost KV pair is popped out. If its $r > 0.5$, we will copy it to the next available slot of each head in the out-of-window segment. If the out-of-window segment is already full, we will evict the lowest-score entry if it scores higher. A fused Triton kernel for this cache replacement process is implemented to obtain best efficiency. Now there is only one query token, and its relevant KVs are already compacted into a contiguous segment when the SWA segment is full. Hence we can simply utilize the existing highly-efficient `flash_attn_with_kvcache` API that supports `cache_seqlens` during decoding.

**Lazy scoring** We do not need to compute $r_j$ immediately when the computation is possible (i.e. current $n$ reaches $j + R//2$. Instead, we can defer and batchify the LTE computation as long as all the tokens within the receptive field remain in-window. Therefore, we trigger LTE scoring only when uncomputed LTE score $r_j$ requires a token that is about to leave the window, i.e. $N - w = j - R//2$. We then rotate the buffer to move *next_ptr* to 0, and compute the LTE score of tokens from $N - w + R//2$ to $N - R//2$ altogether. Another problem is that LTE consumes pre-RoPE keys, while KV cache stores post-RoPE keys, hence we need to undo RoPE on cached keys by applying the corresponding inverse rotation. With RoPE implemented using cached sinusoids, this is equivalent to applying RoPE on the keys with negated sine values at offset $N - w$. With this trick, LTE scoring is carried out only every $W - R$ steps in parallel with minimal amortized overhead.

### 3.2 HYBRID MODELING

We use Gated DeltaNet as our baseline and backbone linear attention for laLTE and laNSA, thanks to its relatively strong performance on retrieval tasks. We apply a 1:1 interleaving of linear attention and alternative token mixers layers, following the Gated DeltaNet + SWA design by (Yang et al., 2024b) known as Gated DeltaNet-H1 for a simplified comparison. We use $w = 1024$ for SWA and $w = 768$ for LTE. Although the LTE cache size is capped to $b = 512$, the observed per-head average retained counts is around 256, so their effective total memory capacity is comparable.

## 4 EMPIRICAL STUDIES

We evaluate hybrid linear-attention models that interleave Gated DeltaNet (GDN) with various mixers from the hierarchy in Figure 1: sliding-window attention (GDN+SWA), learnable token eviction (laLTE), native sparse attention (laNSA), full-attention (GDN+Attn.), plus several ablated variants of LTE. We compare against pure GDN, NSA, and a strong transformer (Transf.++) following Gu & Dao (2024). Given limited computational resources available, we train 0.4B- and 1.4B-sized models on 10B and 30B tokens from FineWeb-Edu (Penedo et al., 2024), following Yang et al. (2024b). We implement LTE layers, hybrid models, as well as inference kernels for LTE and NSA on top of Flash Linear Attention (Yang & Zhang, 2024). Below we leverage both short-context and long-context/retrieval-intensive benchmarks to assess models above and to demonstrate the effectiveness of our laLTE and laNSA approaches, and compare the prefilling and decoding efficiency. More details are available in Appendix A. Additional results on KV retention rates are given in Appendix B.

Table 1: Results (perplexity and accuracy%) on short-context tasks. Best numbers are bolded.

| Model | Wiki. ppl↓ | LMB ppl↓ | LMB acc↑ | PIQA acc↑ | Hella acc_n↑ | Winog. acc↑ | ArcE acc↑ | ArcC acc_n↑ | BoolQ acc↑ | SciQ acc↑ | SIQA acc↑ | Avg.↑ |
|---|---|---|---|---|---|---|---|---|---|---|---|---|
| | | | | | 0.4B Params., 10B Tokens | | | | | | | |
| GDN | 27.56 | 36.06 | 31.9 | 66.4 | 39.1 | 50.9 | 56.8 | 25.9 | 58.4 | 71.0 | 37.5 | 48.7 |
| GDN+SWA | 26.99 | 33.87 | 33.7 | 66.4 | 39.6 | 51.5 | 56.4 | 27.9 | 60.2 | 72.4 | 38.5 | 49.6 |
| laLTE | 26.65 | **30.03** | 34.4 | 66.1 | **40.2** | **52.8** | 57.2 | 26.3 | **61.8** | **75.6** | 38.1 | **50.3** |
| laNSA | 26.84 | 36.19 | 33.2 | 66.1 | 39.9 | 51.8 | **58.0** | **28.3** | 61.6 | 73.1 | 37.7 | 50.0 |
| GDN+Attn. | **26.61** | 33.50 | **34.6** | **66.5** | 39.8 | 51.6 | 57.4 | 27.0 | 55.8 | 74.9 | 38.2 | 49.5 |
| NSA | 27.70 | 39.33 | 33.5 | 66.4 | 38.7 | 50.8 | 55.8 | 28.1 | 59.7 | 69.4 | 37.3 | 48.8 |
| Transf.++ | 27.97 | 40.36 | 32.8 | 65.9 | 38.3 | 51.1 | 56.8 | 28.0 | 61.2 | 71.0 | **38.6** | 49.3 |
| | | | | | 1.4B Params., 30B Tokens | | | | | | | |
| GDN | 18.31 | 15.27 | 43.2 | 70.9 | 52.4 | 54.6 | 67.8 | 34.4 | 58.0 | **83.9** | 40.4 | 56.2 |
| GDN+SWA | 18.53 | 14.77 | 43.7 | 70.4 | 51.7 | 55.0 | 66.8 | 35.2 | 61.4 | 81.5 | 40.2 | 56.2 |
| laLTE | **17.99** | 14.79 | 43.9 | 71.1 | **54.0** | **55.2** | **68.4** | **35.8** | 60.6 | 83.4 | 39.4 | **56.9** |
| laNSA | 18.16 | **14.73** | **44.9** | **71.2** | 52.4 | 54.6 | 67.3 | 35.5 | 59.9 | 82.4 | **40.8** | 56.6 |
| GDN+Attn. | 18.68 | 16.10 | 43.2 | 69.6 | 51.0 | 53.5 | 66.3 | 32.8 | 53.8 | 81.5 | 40.1 | 54.6 |
| NSA | 19.53 | 18.48 | 42.2 | 68.8 | 50.1 | 53.0 | 65.6 | 34.0 | 59.1 | 80.5 | 39.1 | 54.7 |
| Transf.++ | 20.68 | 20.17 | 39.1 | 68.1 | 47.2 | 52.0 | 63.8 | 33.5 | **62.5** | 80.5 | 38.8 | 54.0 |

## 4.1 SHORT-CONTEXT BENCHMARKS

We first evaluate zero-shot commonsense tasks using LM-evaluation-harness (Gao et al., 2024). Given short inputs (generally shorter than the SWA window), we do not expect denser attention to confer any advantage. Thus performance should be broadly similar across models, and differences most likely reflect training quality rather than access to distant tokens. This is confirmed by results in Table 1. At 1.4B, GDN slightly outperforms Transformer++, while NSA and GDN+Attn. fall in between; the ordering is not consistent at 0.4B. GDN+SWA performs similar or slightly better, while adding LTE or NSA further yields small gains. Overall, the gaps are small and can only corroborate Yang et al. (2024b)'s observations that linear attention and hybrid models (including our laLTE and laNSA) are comparable to or slightly better than transformers on short-context tasks.

## 4.2 RETRIEVAL-INTENSIVE BENCHMARKS

We then assess on single needle-in-a-haystack (S-NIAH) from RULER (Hsieh et al., 2024) and the retrieval-intensive EVAPORATE suite (Arora et al., 2024b). S-NIAH spans tasks from retrieving numbers in synthetic texts (S1) to the rather challenging extraction of long UUIDs from real essays (S3). We evaluate contexts of 1K–4K tokens, where most models attain non-trivial accuracy. EVAP-ORATE comprises realistic QA-style retrieval over passages up to 4K tokens, which is challenging for linear-attention models and is widely used in the area. With the demand to retrieve from distant past, we expect that hybrid models with more direct and accurate access to past tokens will have better performance, for which laLTE and laNSA have an edge when time and/or space complexity is constrained. Nevertheless, we acknowledge that access to past tokens alone does not always translate into successful retrieval, higher computational complexity does not monotonically improve accuracy, and the actual outcomes can be task-dependent as observed by Wang et al. (2025).

Results in Table 2 and Table 3 confirm the effectiveness of hybrid models, laLTE and laNSA in particular. Specifically, pure GDN excels on S-NIAH; at 1.4B it approaches full transformers, echo-ing Yang et al. (2024b). However, GDN lags on EVAPORATE. GDN+SWA considerably improves on it, but is much weaker on S-NIAH, especially beyond its 1K window size and at 1.4B. At the cost of efficiency, GDN+Attn. consistently achieves best EVAPORATE results across scales, sur-passing full transformers, and performs comparably to full transformers and pure GDN on S-NIAH. As for our models, despite similar time and space complexity, laLTE outperforms pure GDN and GDN+SWA on both suites, and approaches GDN+Attn. on S-NIAH at 1.4B. This confirms laLTE's effectiveness and versatility for long-context retrieval under strict complexity budgets. With full history kept in memory, laNSA further attains the best results among linear-time models on EVAP-ORATE, approaches full transformers on 0.4B scale S-NIAH, and outperforms pure NSA. However it trails full transformer and GDN+Attn. on EVAPORATE and weaker on 1.4B S-NIAH, despite theoretical all-history access. It also has lower performance than laLTE on retrieval from shorter context (e.g. S3-1K), possibly due to conflict between different attention branches (Hu et al., 2025). Notably, NSA requires at least 16 group size for grouped-query attention. In our models with rel-

Table 2: Results (%) on single needle-in-a-haystack benchmarks from the RULER suite. Numbers higher than 80% are bolded.

| Model | S1-1K | S1-2K | S1-4K | S2-1K | S2-2K | S2-4K | S3-1K | S3-2K | S3-4K | Avg. |
|---|---|---|---|---|---|---|---|---|---|---|
| 0.4B Params., 10B Tokens | | | | | | | | | | |
| GDN | **99.8** | **92.8** | 45.2 | **100.0** | **94.8** | 32.4 | 0.2 | 0.2 | 0.0 | 51.7 |
| GDN+SWA | **100.0** | 52.0 | 27.0 | 61.2 | 67.2 | 27.4 | 75.4 | 64.8 | 17.4 | 54.7 |
| laLTE | **99.8** | **86.8** | 36.4 | **99.8** | 74.2 | 25.0 | **87.2** | 42.8 | 17.8 | 63.3 |
| laNSA | **100.0** | **100.0** | 65.0 | **88.6** | **100.0** | 37.6 | **97.6** | 73.0 | 11.8 | 74.8 |
| GDN+Attn. | **100.0** | **100.0** | 77.0 | **100.0** | **99.6** | 52.6 | **95.4** | **87.0** | 12.2 | 80.4 |
| NSA | **100.0** | **98.8** | 46.6 | **100.0** | **95.8** | 36.4 | **94.4** | 64.6 | 18.6 | 72.8 |
| Transf.++ | **100.0** | **99.6** | 50.8 | **100.0** | **100.0** | 55.4 | 62.8 | 71.4 | 35.8 | 75.1 |
| LTE-MLP | **95.6** | 50.0 | 23.0 | **100.0** | 56.6 | 24.4 | 25.4 | 32.0 | 16.2 | 47.0 |
| Pure LTE | **94.4** | 37.8 | 21.4 | 74.2 | 54.6 | 22.6 | **85.8** | 37.6 | 15.0 | 49.3 |
| 1.4B Params, 30B Tokens | | | | | | | | | | |
| GDN | **100.0** | **100.0** | **100.0** | **93.6** | **99.0** | **88.0** | **93.0** | **77.6** | 49.0 | 88.9 |
| GDN+SWA | **100.0** | 52.0 | 29.6 | **93.4** | **83.2** | 34.2 | **92.8** | 57.0 | 19.4 | 62.4 |
| laLTE | **100.0** | **99.2** | **95.0** | **100.0** | **98.0** | **81.4** | **85.4** | 55.6 | 33.2 | 83.1 |
| laNSA | **100.0** | **99.8** | 57.8 | **80.6** | **100.0** | 68.2 | 60.4 | **94.2** | 34.0 | 77.2 |
| GDN+Attn. | **99.8** | **100.0** | 78.4 | **100.0** | **100.0** | **90.6** | **83.8** | 71.0 | 58.0 | 86.8 |
| NSA | **100.0** | **93.8** | 36.2 | **100.0** | **99.4** | 40.2 | **97.8** | **78.6** | 26.8 | 74.8 |
| Transf.++ | **100.0** | **100.0** | **98.4** | **100.0** | **100.0** | **85.6** | **89.4** | **84.2** | 64.8 | 91.4 |
| TOVA | **99.8** | **81.6** | 41.0 | **100.0** | 11.4 | 8.0 | **83.8** | 0.0 | 0.8 | 47.4 |
| Unif.+SWA | **94.2** | 37.6 | 20.2 | **100.0** | 54.4 | 24.2 | **84.8** | 37.2 | 23.0 | 52.8 |

Table 3: Results (%) on recall-intensive EVAPORATE tasks. Best linear-time results are bolded.

| Model | FDA | SWDE | NQ | SQUAD | TQA | DROP | Avg. |
|---|---|---|---|---|---|---|---|
| 0.4B Params., 10B Tokens | | | | | | | |
| GDN | 4.17 | 8.37 | 10.90 | 27.95 | 47.69 | 17.73 | 19.47 |
| GDN+SWA | 5.08 | 18.27 | 9.34 | **34.52** | 49.11 | 20.51 | 22.81 |
| laLTE | 15.52 | 19.26 | **11.15** | 32.98 | 48.99 | **23.67** | 25.26 |
| laNSA | **17.33** | 23.13 | 10.71 | 31.60 | **49.17** | 21.56 | **25.58** |
| GDN+Attn. | 29.31 | 26.19 | 11.53 | 30.53 | 46.92 | 18.69 | 27.19 |
| NSA | 7.99 | **26.28** | 10.01 | 33.91 | 47.16 | 19.74 | 24.18 |
| Transf.++ | 28.58 | 26.46 | 11.09 | 31.94 | 46.68 | 17.30 | 27.01 |
| LTE-MLP | 12.70 | 18.63 | 11.15 | 31.27 | 49.76 | 20.17 | 23.95 |
| Pure LTE | 14.25 | 18.72 | 7.44 | 33.31 | 46.15 | 19.45 | 23.22 |
| 1.4B Params, 30B Tokens | | | | | | | |
| GDN | 20.96 | 25.83 | **17.96** | 35.02 | 56.46 | 20.75 | 29.50 |
| GDN+SWA | 26.68 | 27.99 | 16.53 | 41.09 | 58.95 | **23.29** | 32.42 |
| laLTE | 28.68 | 28.98 | 17.68 | **43.00** | **60.60** | 22.62 | 33.59 |
| laNSA | **31.85** | **36.99** | 17.20 | 38.07 | 58.06 | 20.60 | **33.80** |
| GDN+Attn. | 46.64 | 41.13 | 18.09 | 40.21 | 58.47 | 22.57 | 37.85 |
| NSA | 25.59 | 34.83 | 13.91 | 40.65 | 56.28 | 23.05 | 32.38 |
| Transf.++ | 38.84 | 34.92 | 16.19 | 37.60 | 55.92 | 21.66 | 34.19 |
| TOVA | 20.96 | 30.78 | 13.49 | 40.18 | 58.59 | 22.62 | 31.10 |
| Unif.+SWA | 15.97 | 24.21 | 11.82 | 22.32 | 57.17 | 15.81 | 24.55 |

atively small hidden sizes and head counts, NSA is rendered closer to multi-query attention, which can be a performance bottleneck and a factor of instability.

We further ablate laLTE against GDN+following token mixers with constant time and space complexity: **LTE-MLP** replaces CNN with a 2-layer MLP with no context access (especially no future access); **Pure LTE** is a pure transformer with all layers sparsified by LTE; **Unif.+SWA** retains out-of-window tokens at a uniform stride; **TOVA** is one of the state-of-the-art eviction heuristics based on accumulated attention scores (Oren et al., 2024); the latter two are training-free. Although LTE-MLP and TOVA achieve moderate results on EVAPORATE, better than pure GDN, all of them

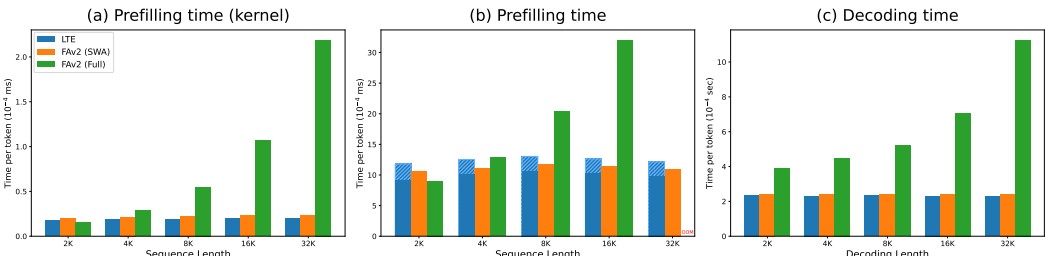

Figure 4: Comparing time costs between LTE and SWA or full attention with Flash Attention-2, on (a) attention kernels prefilling with artificial KVs; and hybrid layers with real inputs during (b) prefilling and (c) decoding with a 4K-token prompt, using our 1.4B models with batch size=32. Shadowed part of the LTE bar in (b) indicate time spent on scoring. Note that decoding takes much longer time than prefilling. Overall, LTE and SWA take similar time, much lower than full attention.

underperform laLTE, especially on S-NIAH; while Unif.+SWA performs much worse than all other models. This confirms the importance of an accurate eviction rule, and underscores the effectiveness of our design of contextualized and learnable eviction. The observation is similar on Pure LTE, which supports our choice to combine sparse attention with linear attention to compensate for the evicted memory. To summarize, all our results confirm the importance of direct and accurate access of past tokens on retrieval-intensive tasks. From GDN, GDN+SWA/laLTE, laNSA, to GDN+Attn., more complete direct access to past tokens generally allows better long-context performance, though task-level outcomes vary; while our laLTE proves its strength for accurate access under rigorous time and space complexity constraints.

### 4.3 EFFICIENCY

We then benchmark the actual computational efficiency of our LTE approach. All results are measured on a single H100-SXM GPU following our 1.4B configuration with batch size=32. Figure 4 (a) shows the runtime of prefilling kernels at different sequence lengths, compared with SWA and full attention using the highly-optimized FlashAttention-2 (Dao, 2024) kernels. As for LTE we use an artificial half-full out-of-window KV-cache, with 1024 tokens identical to the SWA window size. Our SWA+KV-sparse attention kernel exhibits a linear-growing runtime comparable to SWA, and much faster than full attention. While LTE depends on real context for token eviction decision; many layers and heads retain only a few tokens, but the LTE scoring and caching introduce some additional overheads. We therefore also measure the prefilling and decoding time of our actual 1.4B models on real inputs, implemented based on Flash Linear Attention. To better isolate the speed of our targets, we only measure the time costs of hybrid layers (i.e. excluding other layers like GDN). Results shown in Figure 4 (b)-(c) similarly confirm the efficiency of our implementation. Scoring causes some overheads during prefilling (indicated by the shadowed part), which are largely compensated for by faster attention computation. Thanks to lazy scoring, the overhead becomes ignorable and LTE runs slightly faster than SWA during the decoding stage, which dominates inference time. It is further noteworthy that LTE uses a constant-size KV-cache equal to 1280 tokens, i.e. 2.5MiB per sample per layer under our 1.4B configuration. Compared to full KV cache under longer context, this allows larger batch sizes and higher token throughput with fixed size memory. Taken together, these results support adopting LTE in lieu of SWA for better performance without much efficiency degradation.

### 5 CONCLUSION

We study hybrid models with a series of token mixers to improve the retrieval capability of linear attention methods, trying to trade off between time/space complexity and accuracy. We further propose two approaches with different efficiency: laNSA to interleave query-aware native sparse attention of sub-quadratic time complexity when memory capacity permits, and laLTE for token eviction using a learnable contextualized module that maintains constant space complexity and supports an efficient decoding algorithm. Empirical results confirm the effectiveness of our approaches.

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

## A   IMPLEMENTATION DETAILS

We train our models using Flame (Zhang & Yang, 2025) and refer to their training recipes, while adapted to our computational resources available. All the models are implemented based on Flash Linear Attention (FLA). Particularly, FLA implements the prefilling kernel for NSA, and we develop the decoding kernel based on it. It is noteworthy that NSA implemented by FLA is simplified by directly using mean pooling to produce the compressed $k_m^B$ and $v_m^B$. We also develop the LTE prefilling kernel upon the FLA attention kernel. Both the 0.4B and 1.4B models have 24 layers, with hidden size = 1024 and 2048 respectively. As for GDN layers, we use 8 heads and expand_v = 1. As

for attention layers, we set number of heads $H_q = H_{kv} = 16$ at 0.4B, and grouped-query attention (Ainslie et al., 2023a) with $H_q = 32, H_{kv} = 8$ at 1.4B. Similar configurations are also adopted by Wang et al. (2025). The exception is that NSA requires at least 16 group size due to hardware constraints, hence we use $H_q = 32, H_{kv} = 2$. Token embeddings are tied on 0.4B models. The LTE module consists of a 3-layer 1D grouped convolution stack, using kernel size = 3 and dilation = 2, yielding a receptive field $R = 13$. Each layer halves the channel width. Each layer is followed by a Swish activation and a dropout layer with $p = 0.1$. It then passes linear projection per head implemented as grouped convolution with kernel size = 1, and produces the retention score $r$ after sigmoid. We implement directly with Conv2D to avoid tensor manipulation overheads. The number of groups equals the number of KV heads to process each head independently and efficiently. The LTE module will add $\sim 1.7\%$ and $\sim 0.7\%$ parameters on 0.4B and 1.4B models respectively.

We use the AdamW optimizer with learning rate = 3e-4 under cosine decay. At 0.4B/1.4B, we use global batch sizes of 0.5M/1.0M tokens, context length 4096, for 20480/30720 steps with 1024/512 warmup steps. As for LTE training, we initialize the regularization weight matrix $\lambda$ to 1e-9, and update it via a feedback loop every $u = 32$ steps for stability, following the pseudocode given in Algorithm 1. The update is based on the exponential moving average $\bar{c}$ of the retention count $c$, with coefficient $\alpha_{ema} = 2/(1 + u/2)$ to approximate the average within the update cycle.

---

**Algorithm 1** Pseudo-code for the LTE training loop.

---

1: $\lambda_: \leftarrow 10^{-9}$      ▷ Initialize $\lambda$ matrix (shaped [#layers, #heads])
2: **for** step $\leftarrow 0$ to $T$ **do**      ▷ $T$ is total training steps
3:      $r, c, \ell \leftarrow$ model(fetch_inputs())      ▷ Retention score, retention count, and LM loss
4:      $\ell \leftarrow \ell + \lambda \cdot \text{ReLU}(r)$      ▷ Augment loss with L1 penalty on $r$
5:      optimizer.step()
6:      $\bar{c} \leftarrow \alpha_{ema} \cdot \bar{c} + (1 - \alpha_{ema}) \cdot c$      ▷ Exponential moving average of retained token counts
7:      **if** step mod $u = 0$ **then**      ▷ Every $u$ steps, update $\lambda$
8:          Synchronize $\bar{c}$ across devices
9:          $s \leftarrow \mathbf{1}(\bar{c} > L) - \mathbf{1}(\bar{c} < 0.95 \cdot L)$      ▷ Direction to adjust $\lambda$
10:         $\lambda \leftarrow \lambda \cdot \alpha^s$
11:        $\lambda \leftarrow \min(\lambda, 1.0)$      ▷ Clamp $\lambda$ to at most 1.0
12:        $\lambda[\lambda < 10^{-9}] \leftarrow 0$      ▷ Eliminate near-zero values
13:        $\lambda[\bar{c} > L \ \& \ \lambda = 0] \leftarrow 10^{-9}$      ▷ Re-enable penalty when too many retained
14:      **end if**
15: **end for**

---

During LTE decoding, we directly use the `flash_attention_kv_cache` API by Flash Attention, which supports different past KV cache length per sample in a batch by passing in `cache_seqlens`. Since we have different cache length per head, we merge the dimensions for KV head and batch size, leaving a KV cache with the number of query heads equals to the GQA group size. As for evaluation, we use LM-eval-harness for short-context and RULER evaluations, and leverage the reference implementation by Arora et al. (2024c) for EVAPORATE benchmarks. We follow their input truncation scheme, but only truncate the input length to 4K tokens when it exceeds 4K to better reflect long-context performance. Inference is carried out using greedy decoding in FP32 for stability, except that GDN and Flash Attention kernels support FP16 only, hence we perform all token-mixing ops in FP16 for parity. Training-free sparse attention mechanisms including Unif.+SWA and TOVA are evaluated by applying them to pretrained GDN + Attn. We use an identical 1024 total cache size for both of them. For compute-cost measurements, we report the total elapsed time between the CUDA event pairs at layer entry after a full warm-up phase; for prefilling we average the middle three of five runs. Samples from the training dataset are used as inputs.

## B RETENTION PATTERN

Figure 5 shows the average retention rate per head per layer from the out-of-SWA part of 16 samples drawn from the training data produced by our 1.4B laLTE model. We find that retention rates are highly varied across layers and heads, while the higher layers are often denser. This is similar to findings in other works, e.g. Łańcucki et al. (2025), except that dense heads do not present in lowest

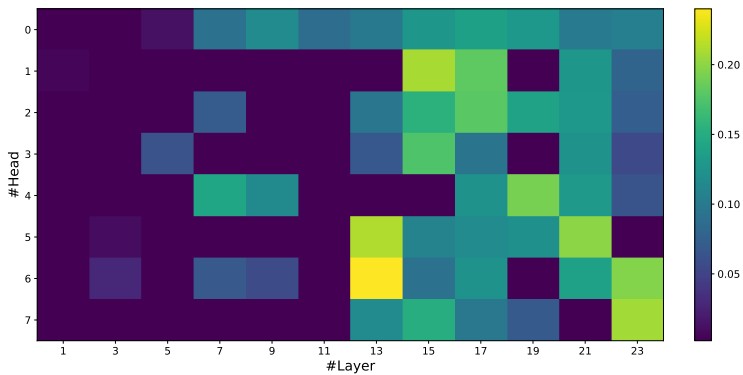

Figure 5: Average retention rate per head per layer produced by a laLTE model.

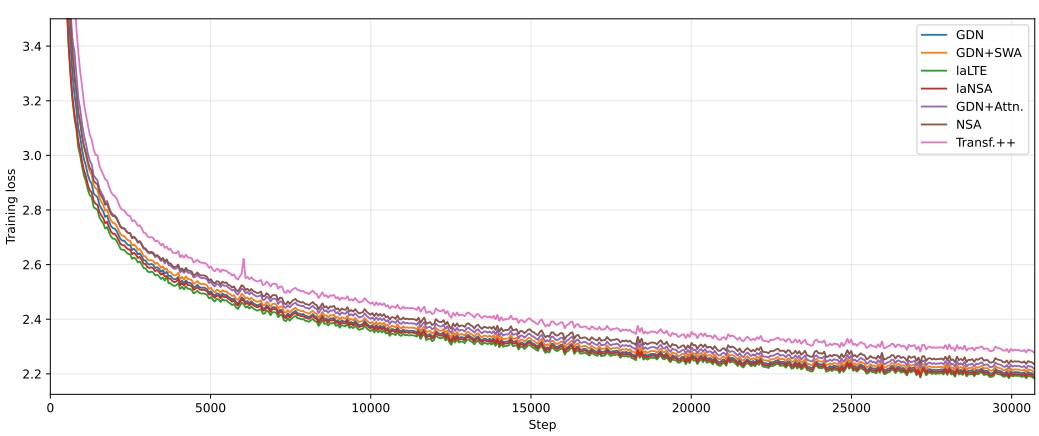

Figure 6: Training loss curve of 1.4B models.

layers; The interleaved GDN layers may have a role in it. The observation also supports the necessity of having a fine-grained cache budget different from head to head and from layer to layer.

## C LOSS CURVES

Figure 6 shows the loss curves of our 1.4B models, which match our expectation, observations in previous works, and short-context evaluations: both NSA and GDN have lower perplexity than pure transformers, while hybrid models (specifically, laLTE and laNSA) will further improve the results, though the training loss doesn't indicate long-context capabilities.

