# OpenReview forum: "Alleviating Forgetfulness of Linear Attention by Hybrid Sparse Attention and Contextualized Learnable Token Eviction"
_ICLR.cc/2026/Conference — ICLR 2026 Conference Withdrawn Submission_

### Official Review · Reviewer_Hr9r · 2025-10-23

**Soundness:** 2
**Presentation:** 2
**Contribution:** 2
**Rating:** 2
**Confidence:** 3

**Summary:**

The authors propose a form of learned token eviction which uses a CNN to learn the token eviction scores within a sliding window. The proposed method requires training from scratch.

**Strengths:**

- Sparse Attention is an important topic for long context models as the attention operation is quadratic
- The authors utilize many recent works as a motivation and backbone for their method.

**Weaknesses:**

- L165: The probing step remains quadratic due to the constant setting of the block size $M$. As this is part of the attention computation as a precursor operation, I don't think you claim that the attention computation is a constant $MK$.

- How can the method presented in figure 2 ever learn to retain a token in a simple task such as needle-in-a-haystack? For example, datasets like RULER have some tasks where the retrieval target can be any number of key value pairs which is not known until the query is given at the end of the prompt? How could this scheme know what to retain in this situation?

- Figure 3 says the out of window KV's have a capped capacity, but it looks like the second row extends to all previous KV's. Shouldn't this be capped instead of storing all of them?

- I don't quite understand equation 6. If the mask was applied, meaning that there was no computation in the forward pass for $v^\prime_{j,h}$, as this computation was skipped. Is the last term on the RHS supposed to be $\langle v_{j,h}, \frac{\partial \mathcal{L}}{\partial v^\prime_{j,h} \rangle$?

- L301 states that unimportant tokens will receive negative gradients as a matter of fact. How can you be so sure of this?

- Table 2 is very hard to make sense of with all numbers above 70% being bolded.

- This is supposed to be for linear (and therefore efficient) attention, but there is no latency comparison between models.

---

Overall, I find the method and the results to be underwhelming at best. The results seem mixed and inconclusive between methods which makes it hard to pin down the exact strengths of the method. As there is no latency comparison, readers have no idea of the overall efficiency compared to the other baseline methods. It is also hard to see the exact benefit of the learned token eviction when it is not compared to a simple baseline such as aggregated attention scores within the window.

---

## Minor

L107: stemm --> stemming?

L130: "tokens can be self-evidently crucial" --> I find "self-evidently" to be problematic here. If they were self evidently crucial, then they would be easy to identify and retain. But in the case of a passkey, it is not self evident that they are crucial for the task without perfect knowledge of the query which will be asked. I think these words should be rephrased.

**Questions:**

- RULER was only considered for NIAH single tasks up to 4K. Can you extend this to the more complicated tasks with longer contexts? The model that uses NSA should be capable of handling this at least, right?

- Could it be possible that getting ride of the LTE module and only tracking token scores could lead to a similar or better eviction policy?

---

> ### Author Response · Authors · 2025-11-18
>
> Thanks for the review, and we address the concerns below.
>
> # Efficiency
>
> We added the actual latency measurements to the updated manuscript in Fig. 4 and Sec. 4.3. To sum up, the computational cost of LTE is similar to FlashAttention+SWA and much lower than full attention, despite LTE’s more sophisticated attention pattern and improved performance, thanks to our dedicated caching, prefilling, and decoding scheme.
>
> # Baselines & ablations
>
> The manuscript has already reported results on several ablated variants in Tabs. 2 and 3 to confirm the advantage of our design, as described in Sec. 4. Among them is TOVA, a representative of the state-of-the-art training-free token eviction approaches based on token attention scores. The results demonstrate the importance of a trained (instead of heuristic-based) token eviction approach, as carried out by LTE.
>
> # Other questions
>
> > L165: The probing step remains quadratic ... I don't think you claim that the attention computation is a constant $MK$.
>
> Indeed, the probing step remains quadratic, though in a much more lightweight fashion. Accordingly, Fig. 1 specifies that attention computation takes constant time *given selected KVs*. This is an inherent weakness of query-aware sparse attention, and we have updated the manuscript to further emphasize this point.
>
> > How can the method presented in figure 2 ever learn...How could this scheme know what to retain in this situation?
>
> > L130: "tokens can be self-evidently crucial" --> I find "self-evidently" to be problematic here.
>
> We agree that any method that maintains a constant memory cost without access to future queries will necessarily struggle on needle-in-a-haystack–style scenarios, where keys that appeared totally irrelevant in the past are abruptly queried at the end. However, a rich body of previous token-eviction heuristics that simply retain tokens with high local attention scores has already achieved good results on many retrieval tasks. This suggests that the model can identify words that are *typically* useful in the future (as opposed to, e.g., function words). Our method builds on this line of work by introducing a learnable component that predicts *typically* useful tokens even when they do not receive high local attention scores. By training on signals from the entire context, the model learns patterns of such typicality, for example, category of film in its infobox (SWDE) or an unusual pass key in the middle of an essay (S-NIAH-2) is likely to be important. We have updated the manuscript to clarify this intuition and to avoid the potentially confusing phrasing “self-evidently crucial”.
>
> > Figure 3 says the out of window KV's have a capped capacity, but it looks like the second row extends to all previous KV's. Shouldn't this be capped instead of storing all of them?
>
> Figure 3 illustrates the KV cache after eviction, not the original KVs. The second row actually indicates a head with a fully filled cache.
>
> > I don't quite understand equation 6. If the mask was applied, meaning that there was no computation in the forward pass for v' \_{j,h}, as this computation was skipped. Is the last term on the RHS supposed to be $\langle v_{j,h}, \frac{\partial \mathcal{L}}{\partial v^\prime_{j,h} \rangle$?
>
> > L301 states that unimportant tokens will receive negative gradients as a matter of fact. How can you be so sure of this?
>
> The $m_{j,h}$ is a conceptual all-one mask applied to the values, distinct from the mask actually used in the attention computation. In practice, all $v_{j,h}$ will receive some gradients from the SWA part even if it will be evicted. The gradient computation is equivalent to a straight-through estimator, while the mask perspective allows us to view it internally: when the all-one conceptual mask $m_{j,h}$ receives negative gradients that drive it to zero, the gradient from the loss is actually driving the value vector $v_{j,h}$ to zero, which indicates that the token is unimportant. The manuscript has been updated to clarified this.
>
> > RULER was only considered for NIAH single tasks up to 4K. Can you extend this to the more complicated tasks with longer contexts? The model that uses NSA should be capable of handling this at least, right?
>
> We report the results up to 4K tokens because the models suffer from much lower performance in longer contexts, and hence the numbers are less meaningful. Also, the EVAPORATE suite has already covered a broader range of tasks more complicated than the synthetic RULER S-NIAH. It is noteworthy that, despite access to the full history, NSA or even full-attention models are not guaranteed to be able to precisely retrieve everything in the history, as already found in our experiments.
>
> > Table 2 is very hard to make sense of with all numbers above 70% being bolded.
>
> We have increased the threshold to 80% to make the table easier to read. Other typos have also been fixed.

---

> > ### Comment · Reviewer_Hr9r · 2025-11-25
> >
> > Thank you for your responses. In general, I still find the results presented in table 2 to be confusing. Additionally, I am not convinced that the CNN learning an eviction strategy is the right way to go given that it opens up many new concerns about generalization from training to test time and if the learned eviction strategy will generalize.
> >
> > After revision, I still find it hard to pin down the exact benefits of the method and therefore I will maintain my current score.

---

> > > ### Author Response · Authors · 2025-11-25
> > >
> > > Thanks for the comment. We are happy to clarify any remaining confusion about our results, and to explain the motivation behind our overall design.
> > >
> > > * **Rule-based token eviction** (query-agnostic) has been shown effective in various prior works: even without access to future queries, one can identify KV entries that are *typically* useful across a range of downstream queries.
> > > * **A learned eviction module** enables more flexible, input-dependent decisions, consistent with the general deep learning principle of learning from data rather than using hand-crafted heuristics.
> > > * **Using local context improves eviction decisions.** Compared to eviction based solely on a single token’s feature, mixing information from neighboring tokens provides additional contextual cues that can better predict future utility. We adopt ShortConv for this purpose, which is a standard component in current linear-attention models (e.g., Mamba, RWKV, DeltaNet) to enhance local token mixing, and has been found beneficial even in standard attention (e.g., [Multi-Token Attention](https://openreview.net/forum?id=Z3L35tQTEg)). Particularly, a concurrent work ([FlashMoBA](https://www.arxiv.org/abs/2511.11571)) shows that applying ShortConv to keys is particularly effective for accurate KV selection in query-aware sparse attention, which echoes our design.
> > >
> > > Overall, our CNN-based design is not ad-hoc; it follows established principles and aligns with emerging evidence in recent architecture designs. These claimed benefits are also supported by our ablations (LTE vs. TOVA; CNN vs. MLP).

---

### Official Review · Reviewer_LAgD · 2025-10-31

**Soundness:** 2
**Presentation:** 4
**Contribution:** 2
**Rating:** 2
**Confidence:** 4

**Summary:**

This paper proposes the Learned Token Eviction (LTE) algorithm. The authors combine LTE with linear attention, specifically sliding window attention, and refer to it as laLTE.

**Strengths:**

- The method is simple to implement.

**Weaknesses:**

## Lacks of novelties
The method is somewhat akin to combining existing elements.

## Lacks of efficiency analysis

Even if the method is claimed as linear complexity, the linear complexity does not always mean faster and efficient than flash attention. The author lacks a critical analysis of efficiency in real-world hardware. Any latency seconds were not reported.

Especially about CNN, the latency analysis is really crucial, since the small size of Conv operation is known to be slower than normal vector-matrix and matrix-matrix operations. I want to hear the answer and a detailed analysis of the following things:
 - Really need CNN?
 - How much does CNN slow down in wall clock latency? Please compare it to HW efficient alternatives (Mamba, LightningAttention2)

## Eviction is still a critical problem in a multi-turn request scenario, especially for tool calling

Since the KV cache is more and more critical in an agentic AI scenario, we cannot drop the KV cache without a precise study.

 - Is there any analysis about tool callings?
 - What is the agentic LLM performance?

## Where is the training curve?
 - I cannot be sure this model is sufficiently trained or scalable

## Lacks of some strong performance baselines:
- Mamba2
- Lightening attention https://github.com/MiniMax-AI/MiniMax-M1

I think we need to include such alternatives to build a competitive method for deployable methods.

**Questions:**

## Typo
- line 259: se -> sequence (?)

## Questions
- How does the inverse rotation affect the precision errors? on fp8? fp16? fp4?
- Table 2, you must put the latency.

---

> ### Author Response · Authors · 2025-11-18
>
> Thanks for the review, and we address the concerns below.
>
> # Novelty
>
> Previous hybrid models generally alleviate linear attention’s fuzzy memory issue by either hybridizing linear attention with SWA (e.g., Samba [1], Hymba [2] at ICLR 2025) or with full attention (e.g., Jamba [3], Kimi-Linear [4], Qwen3-Next [5]). These methods can also be reduced to “simply mixing existing techniques”, but they are in fact fairly effective and highly influential. More importantly, both types of methods are unable to precisely retrieve distant content without sacrificing time&space complexity. Hence we emphasize that our model design of using sparse (instead of SWA/full) attention to compensate for linear attention’s inherent drawback is already novel, not to mention the proposed contextualized LTE approach.
>
> # Efficiency
>
> We added the actual latency measurements to the updated manuscript in Fig. 4 and Sec. 4.3. To sum up, the computational cost of LTE is similar to FlashAttention+SWA and much lower than full attention, despite LTE’s more sophisticated attention pattern and improved performance, thanks to our dedicated caching, prefilling, and decoding scheme.
>
> # Tool callings
>
> We follow previous works and report benchmarks for linear and hybrid attention on common language modelling tasks and long-context retrieval tasks. Since we work at the pretraining stage, not the post-training stage, the trained models are not able to follow instructions or call tools. We can nevertheless expect that a pretrained model with stronger retrieval capabilities will perform better on such downstream tasks. It is also noteworthy that DeltaNet can be inherently stronger at state-tracking than full attention [6], hence promising for agentic AI. It is also worth mentioning that linear attention states carry a compressed memory of the whole sequence, even if most of the KV cache is dropped. This may compensate for the loss of memory and may explain the improved performance of laLTE over the pure LTE model without linear attention, as reported in Sec. 4.2.
>
> # Training curves
>
> We added the training curves to Appendix C.
>
> # Baselines
>
> We choose Gated DeltaNet (GDN) as our linear attention baseline, as it represents the state of the art and enjoys much strong retrieval capabilities than earlier methods like Mamba2 as reported by [7], though it is still much weaker than full attention. We also tried Mamba2 in our experiments, but it produced an average of 41.1% and 21.4% accuracy on RULER and EVAPORATE, 50% lower than any of our GDN-based models under identical settings. Lightning Attention is based on an earlier and weaker linear attention mechanism compared to Mamba, using constant memory decay over time. Hence we believe that GDN can represent a sufficiently strong baseline.
>
> The manuscript has already reported results on several ablated variants in Tabs. 2 and 3 to confirm the advantage of our design, as described in Sec. 4 (L481-510). Among them is LTE-MLP, which uses an MLP (i.e. with receptive field=1) instead of CNN for prediction, as in other recent learnable token eviction approaches (though there is a limited body of work on it, as reviewed in Sec 2.1, L133-148). The lower results support our design to use a short CNN to expand the receptive field.
>
> # Precision issues
>
> In FP32 and FP16, the inverse rotation of K leads differences on the order of 1e-8 and 1e-4 respectively, compared to the original K, and causes <0.05% change of selected tokens. Considering that the precision issue will only affect tokens with scores on the borderline, and the eviction decision is made on each layer (hence the error does not accumulate across layers), we do not expect this to create a significant precision issue.
>
> [1] Samba: Simple Hybrid State Space Models for Efficient Unlimited Context Language Modeling. ICLR 2025.
> [2] Hymba: A Hybrid-head Architecture for Small Language Models. ICLR 2025 (Spotlight).
> [3] Jamba: Hybrid Transformer-Mamba Language Models. ICLR 2025.
> [4] Kimi Linear: An Expressive, Efficient Attention Architecture. arXiv: 2510.26692.
> [5] Qwen3-Next: Towards Ultimate Training & Inference Efficiency.
> [6] Unlocking State-Tracking in Linear RNNs Through Negative Eigenvalues. ICLR 2025 (Oral).
> [7] Gated Delta Networks: Improving Mamba2 with Delta Rule. ICLR 2025.

---

### Official Review · Reviewer_uoix · 2025-10-31

**Soundness:** 3
**Presentation:** 3
**Contribution:** 3
**Rating:** 6
**Confidence:** 3

**Summary:**

The paper addresses the recall gap of linear-attention and recurrent LMs by re-introducing targeted access to past tokens without abandoning efficiency. It studies two hybrid designs: (1) laNSA: interleaves linear attention with Native Sparse Attention (NSA) that combines query-aware block probing, a compressed branch, and SWA-style gating. This improves retrieval but keeps an O(N) KV cache. (2) laLTE: interleaves linear attention with Learnable Token Eviction (LTE). A tiny per-token, per-head 1D-CNN predicts whether to retain a KV as it leaves a large SWA window; combined with an attention sink and SWA, this aims for O(1) time and space per step while preserving long-range evidence. Experiments at roughly 0.4B and 1.4B parameters trained on 10B/30B tokens (FineWeb-Edu) evaluate short-context language tasks and long-context retrieval (RULER S-NIAH, EVAPORATE). Both laLTE and laNSA outperform strong linear baselines; laNSA is strongest among linear-time models on EVAPORATE, while laLTE is the best constant-space option and approaches hybrid full-attention variants in some settings.

**Strengths:**

### Novelty:
This work proposes two complementary mixers interleaved with linear attention—NSA for query-aware sparse access over the full past and LTE for learned keep/evict under a strict cache budget; introduces per-token, per-head retention via a tiny 1D-CNN with SWA-enabled look-ahead and an attention sink to maintain near-constant KV memory; provides deployment-minded decoding/KV design (two-segment cache, lazy batched scoring) and frames a clear accuracy–efficiency Pareto frontier (laLTE for constant-space, laNSA for higher accuracy under linear time).

### Differentiation from prior works:
This work moves beyond fixed windows and global/uniform/time-decay heuristics by making head-aware, context-conditioned retention decisions; regains direct token-level access to salient long-range evidence without reverting to O(N²) attention, contrasting with state-space/recurrence approaches that compress history into fixed states; offers a stronger NSA baseline (query-aware probing + compressive branch + SWA gating) that sharpens comparisons.

**Weaknesses:**

### Scale and generality:
Results are limited to 0.4B/1.4B. It is unclear whether the trends hold for larger, modern LLM families (e.g., Qwen2.5/3, DeepSeek) or for multilingual/code models.
### Benchmark breadth:
The evaluation focuses on long-context retrieval. Broader benchmarks commonly used today (e.g., instruction following, math, and code such as AlpacaEval, GSM8K, HumanEval) are absent, making it hard to gauge side effects beyond retrieval.

### Efficiency reporting:
The paper argues constant time/space for laLTE by design, but it does not provide systematic, measured wall-clock throughput and GPU memory usage across mixers (GDN, +SWA, laLTE, laNSA, +full-attention).

### NSA dependency:
laNSA still requires O(N) KV. The trade-off versus laLTE is discussed conceptually; clearer guidance on when laNSA is preferable in practice would help.

**Questions:**

1. Can you report measured GPU memory (GB), effective KV size, tokens/sec, and latency for GDN, GDN+SWA, laLTE, laNSA, and a GDN+full-attention hybrid at 4K and at least 8K contexts on the same hardware?

2. Since laLTE/laNSA are trained modules, could you also include an inference-only comparison on the same GPU (with matched context length and batch size) against training-free efficient attention approaches such as HiP [1]? Reporting retrieval accuracy (e.g., RULER/EVAPORATE), throughput (tokens/s), and peak memory would help clarify whether the additional training cost yields meaningful gains over training-free sparse attention methods.

---
[1] Lee et al. A Training-free Sub-quadratic Cost Transformer Model Serving Framework With Hierarchically Pruned Attention. ICLR 2025.

---

> ### Author Response · Authors · 2025-11-18
>
> Thanks for the review, and we address the concerns below.
>
> # Model scale
>
> Given limited computational resources in an academic setting, we are only able to examine our idea at a relatively small scale; the 0.4B and 1.4B settings are nevertheless also used by other linear-attention works. Also, a model of moderate scale is already valuable for personal and edge use, a scenario particularly suitable for efficient architectures like linear and hybrid attention.
>
> # Benchmark breadth
>
> We follow previous works and report benchmarks for linear and hybrid attention on common language modelling tasks and long-context retrieval tasks. It is noteworthy that we work at the pretraining stage without any post-training, and models at this stage typically lack instruction-following and reasoning capabilities.
>
> # Efficiency
>
> We added the actual latency measurements to the updated manuscript in Fig. 4 and Sec. 4.3. To sum up, the computational cost of LTE is similar to FlashAttention+SWA and much lower than full attention, despite LTE’s more sophisticated attention pattern and improved performance, thanks to our dedicated caching, prefilling, and decoding scheme.
>
> # NSA dependency
>
> We agree that NSA mechanism has multiple limitations: it requires an $O(N)$ KV cache, has higher computational overheads from three separate attention branches and a top-k selection, and may obtain lower results than laLTE when retrieving from relatively short contexts (e.g., S3-1K), possibly due to conflicts between different NSA branches [1]. These observations can be useful for future applications of NSA layers, and the discussions have been included into the manuscript. Nevertheless, these drawbacks of the NSA layer further highlight the strength of laLTE, which is the main contribution of this work.
>
> # Ablation studies
>
> The manuscript has already reported results on several ablated variants in Tabs. 2 and 3 to confirm the advantage of our design, as described in Sec. 4 (L481-510). Among them is TOVA, a representative of the state-of-the-art training-free token eviction approaches. The results demonstrate the importance of a trained (instead of heuristic-based) token eviction approach. While the advantage of NSA over other training-free query-aware sparse methods (including HiP) has already been established in their paper [2]. We have nevertheless added HiP to our review of such query-aware methods in Sec. 2.1.
>
> [1] Optimizing Native Sparse Attention with Latent Attention and Local Global Alternating Strategies. arXiv:2511.00819.
> [2] Native Sparse Attention: Hardware-Aligned and Natively Trainable Sparse Attention. ACL 2025 (Best paper).

---

### Official Review · Reviewer_37ke · 2025-11-01

**Soundness:** 3
**Presentation:** 3
**Contribution:** 2
**Rating:** 2
**Confidence:** 4

**Summary:**

The paper targets the “forgetfulness” of linear-attention models by interleaving Gated DeltaNet layers with stronger token mixers. Two variants are proposed: (1) laLTE, which introduces a learnable token eviction (LTE) module that scores each KV pair per head using a tiny 3-layer 1D CNN with a short receptive field and then retains only a capped number of out-of-window tokens. (2) laNSA, which swaps in Native Sparse Attention (NSA) layers that perform query-aware block, offering more direct access to the past but requiring O(N) KV memory.
The authors position these mixers on a complexity–access hierarchy. Empirically, on EVAPORATE and RULER, the hybrids often outperform pure GDN/GDN+SWA, while full-attention interleaves remain the strongest but are also the most costly.

**Strengths:**

1. Per-head, per-token scoring from short local context with grouped 1D convs is simple, parallel, and adds ~1% params. The design is a constant budget and predictable latency.
2. Results are reported on both synthetic (S-NIAH) and realistic (EVAPORATE) retrieval benchmarks, showing consistent gains over pure GDN/GDN+SWA in many settings.

**Weaknesses:**

1. The novelty is limited. The laNSA component is adopted from prior NSA work, and the overall recipe seems an alternation of existing hybrid attention rather than a fundamentally new mechanism. The idea of LTE also sits close to the broader family of token-eviction methods, making the novelty feel incremental.
2. Evidence does not decisively beat the common practice. The improvements on EVAPORATE are modest averages (e.g., laLTE/laNSA only several points over GDN/GDN+SWA), while interleaving full attention (GDN+Attn.) remains stronger in many settings.
3. No end-to-end latency measurements (prefill & decode) under the claimed constant budgets, nor e2e latency comparisons against strong kernels (e.g., Flash-/Flex-Attention baselines) make it hard to assess the practical gains of LTE beyond proxy complexity.
4. The evaluated models are relatively small (0.4B/1.4B on 10B/30B tokens FineWeb-Edu), which limits the strength of conclusions about scalability to modern LLMs.
5. Ablation study on LTE is insufficient. The paper motivates head-wise independence and a short receptive field, but there is limited analysis on (i) sensitivity to the cap b, window w, and receptive field R; (ii) alternatives to CNN scoring (MLP/other efficient attention predictors).

**Questions:**

1. Could authors report e2e prefill and decode latency vs. GDN, GDN+SWA, and GDN+Attn., all using the same optimized kernels?
2. How does laLTE compare to other recent learnable-eviction or head-aware KV-budgeting methods under a matched training budget?

---

> ### Author Response · Authors · 2025-11-18
>
> Thanks for the review and the concerns are addressed below:
>
> # Novelty
>
> Previous hybrid models generally alleviate linear attention’s fuzzy memory issue by either hybridizing linear attention with SWA (e.g., Samba [1], Hymba [2] at ICLR 2025) or with full attention (e.g., Jamba [3], Kimi-Linear [4], Qwen3-Next [5]). These methods can also be reduced to “simply mixing existing techniques”, but they are in fact fairly effective and highly influential. More importantly, both types of methods are unable to precisely retrieve distant content without sacrificing time&space complexity. Hence we emphasize that our model design of using sparse (instead of SWA/full) attention to compensate for linear attention’s inherent drawback is already novel, not to mention the proposed contextualized LTE approach.
>
> # Efficiency
>
> We added the actual latency measurements to the updated manuscript in Fig. 4 and Sec. 4.3. To sum up, the computational cost of LTE is similar to FlashAttention+SWA and much lower than full attention, despite LTE’s more sophisticated attention pattern and improved performance, thanks to our dedicated caching, prefilling, and decoding scheme.
>
> # Performance
>
> We agree that interleaving full attention remains stronger in many settings, but this comes at the cost of quadratic time, which is what we intend to avoid. Our findings are consistent with our key claim: laLTE can serve as an alternative to the standard GDN+SWA hybrid, with similar efficiency but improved performance, while laNSA may further improve the results on certain tasks.
>
> # Model scale
>
> Given limited computational resources in an academic setting, we are only able to examine our idea at a relatively small scale; the 0.4B and 1.4B settings are nevertheless also used by other linear-attention works. Also, a model of moderate scale is already valuable for personal and edge use, a scenario particularly suitable for efficient architectures like linear and hybrid attention.
>
> # Baselines & ablation studies
>
> The manuscript has already reported results on several ablated variants in Tabs. 2 and 3 to confirm the advantage of our design, as described in Sec. 4 (L481-510), including
> * LTE-MLP, using an MLP (i.e., with receptive field = 1) instead of a CNN for prediction, as in other recent learnable token eviction approaches (though there is a limited body of work on this, as reviewed in Sec. 2.1, L133–148). The results demonstrates the importance of a larger receptive field, which is achieved by our CNN-based prediction.
> * Pure LTE and pure NSA, which demonstrate the importance and synergy of combining linear and sparse attention, possibly by mitigating the risk of missing important tokens in pure sparse attention thanks to linear attention's compressed memory.
> * Uniform + SWA, using a static sparsity pattern, which demonstrates the importance of using dynamic sparse attention like LTE and NSA.
> * TOVA, a representative of the state-of-the-art training-free token eviction approaches, which demonstrates the importance of a trained (instead of heuristic-based) token eviction approach.
>
> [1] Samba: Simple Hybrid State Space Models for Efficient Unlimited Context Language Modeling. ICLR 2025.
> [2] Hymba: A Hybrid-head Architecture for Small Language Models. ICLR 2025 (Spotlight).
> [3] Jamba: Hybrid Transformer-Mamba Language Models. ICLR 2025.
> [4] Kimi Linear: An Expressive, Efficient Attention Architecture. arXiv: 2510.26692.
> [5] Qwen3-Next: Towards Ultimate Training & Inference Efficiency.

---

### Author Response · Authors · 2025-12-03
**Summary of rebuttal**

We thank the reviewers for their comments and suggestions. As most of the reviewers have not provided follow-up responses, we summarize below the main concerns raised and our corresponding revisions and clarifications.

## Wall-clock time measurements

In the revised manuscript, we added end-to-end latency measurements in Fig. 4 and Sec. 4.3. In summary, the computational cost of LTE is roughly identical to the sliding-window attention implemented by FlashAttention-2, and much lower than full attention, thanks to our dedicated caching, prefilling, and decoding scheme. While LTE has much higher accuracy based on its more sophisticated attention pattern compared to SWA, suggesting that LTE is a favorable alternative to SWA within hybrid linear-attention architectures.

## Novelty

Two reviewers question about the novelty of our methods, characterizing hybrid models as "combining existing elements". We would like to emphasize that hybrid modeling is currently a central research direction for linear attention. Recent hybrid models generally alleviate linear attention’s fuzzy memory issue by either hybridizing linear attention with SWA (e.g., Samba [1], Hymba [2] on ICLR 2025) or with full attention (e.g., Jamba [3], Kimi-Linear [4], Qwen3-Next [5]), which are fairly effective and highly influential. However, both types of methods are unable to precisely retrieve distant content without sacrificing time and space complexity. Therefore, our design choice to use sparse (rather than SWA/full) attention to compensate for linear attention’s inherent limitations is itself novel, not to mention the proposed contextualized LTE approach.

## Model scale and benchmark breadth

Several reviewers commented on the scale of models we use, and request for evaluations on a broader set of reasoning and agentic benchmarks. We would like to clarify that, for architectural research on LLM pretraining in an academic setting, we conducted experiments at moderate scales (0.4B and 1.4B parameters). At this pretraining-only stage (i.e., without post-training), none of the models will be able to reliably perform complex reasoning or tool-use tasks. While we have already provided empirical evaluations on language modelling/commonsense and long-context retrieval tasks, sufficient to support our hypotheses. Moreover, models at this scale are also practically valuable for personal and edge deployment, which are scenarios particularly well matched to efficient architectures such as linear and hybrid attention.

## Ablation studies

Several reviewers ask for additional ablations (e.g., alternative scoring methods). We note that the manuscript already reports multiple ablated variants in Tabs. 2 and 3 and discusses them in Sec. 4 (L481–510) to confirm the advantages of our design, including:

* LTE-MLP, using an MLP (i.e., with receptive field = 1) instead of a CNN for prediction, which aligns with some concurrent learnable token eviction approaches; there is a limited body of work on this topic, as reviewed in Sec. 2.1, L133–148. The degraded results demonstrates the importance of contextualized scoring with a larger receptive field, which is enabled by our CNN-based module.
* Pure LTE and pure NSA, which demonstrate the importance and synergy of combining linear and sparse attention, potentially by mitigating the risk of missing important tokens in pure sparse attention thanks to linear attention's compressed memory.
* Uniform + SWA, using a static sparsity pattern; the worse results demonstrate the importance of using dynamic sparse attention like LTE and NSA.
* TOVA, as a representative of the state-of-the-art training-free token eviction approaches; the weaker results support the need for learned (rather than heuristic) token eviction.


[1] Samba: Simple Hybrid State Space Models for Efficient Unlimited Context Language Modeling. ICLR 2025.
[2] Hymba: A Hybrid-head Architecture for Small Language Models. ICLR 2025 (Spotlight).
[3] Jamba: Hybrid Transformer-Mamba Language Models. ICLR 2025.
[4] Kimi Linear: An Expressive, Efficient Attention Architecture. arXiv: 2510.26692.
[5] Qwen3-Next: Towards Ultimate Training & Inference Efficiency.

---

### Note · Authors · 2026-01-06

I have read and agree with the venue's withdrawal policy on behalf of myself and my co-authors.